# Linkage Between the Environment and Individual Resilience to Urban Flooding: A Case Study of Shenzhen, China

**DOI:** 10.3390/ijerph16142559

**Published:** 2019-07-18

**Authors:** Jing Song, Weifeng Li

**Affiliations:** 1Key Laboratory for Environmental and Urban Sciences, School of Urban Planning and Design, Shenzhen Graduate School, Peking University, Shenzhen 518055, China; 2Qianhai Institute for Innovative Research, Shenzhen 518052, China; 3Department of Urban Planning and Design, University of Hong Kong, Hong Kong SAR, China; 4Shenzhen Institute of Research and Innovation, The University of Hong Kong, Shenzhen 518057, China

**Keywords:** individual resilience, human behavior, environmental cognition, urban flooding, hierarchical linear model (HLM)

## Abstract

Resilience is widely accepted as the capacities implemented to manage climate change. Exploring how individual resilience can be enhanced to better prepare residents for natural disasters, such as urban flooding, is therefore necessary. Environmental cognitions that provide psychological and physiological benefits to people by adding motivation to interact with the place are factors influencing people’s resilience-oriented behaviors but have largely been ignored in existing research. As such, this study establishes a framework for the concept of individual resilience to urban flooding. Gongming, a sub-district of Shenzhen, China, is considered the case area wherein individual resilience and its environmental determinants are evaluated. Through hierarchical linear modeling, the environmental determinants of individual resilience at the individual and community levels are identified. At the individual level, the main factors are a few green spaces, low quality of the built environment, mutual distrust and lack of well-being perceived by residents. At the community level, the results suggest that the social environment, particularly its gatedness, is pivotal to individual resilience. This study offers an approach for analyzing factors that limit individual resilience from the environmental perspective, thereby providing a basis for formulating corresponding policy recommendations to effectively improve resilience through urban planning.

## 1. Introduction

Resilience theory provides insights into managing uncertainties in complex socio-ecological systems, especially with respect to dramatic landscape changes [1,2]. City systems are dynamic, self-organizing, and able to withstand disturbances of a certain magnitude [3]. However, with the rising influence of human behavior on a city system, disturbances have become increasingly more complex and changeable in nonlinear ways [4]. Developing measures that can expand the capacity of urban systems, including those of built environment and human beings, to withstand a wide array of external shocks has become an urgent concern. This, in essence, involves the concept of resilience, which refers to abilities of preparedness, planning, absorption, recovery, and adaptation that address uncertain and unpredictable states of flux [5].

As the world is increasingly urbanized, the adverse impacts of dramatic landscape changes are emerging as primary threats to the planet [6]. Urban flooding, a natural disaster that results from landscape changes, has affected many areas around the globe, causing economic damage and human misery, particularly in developing countries, such as China [7]. Statistical data from the Ministry of Housing and Urban-Rural Development show that in China, 62% of 351 cities suffer from inundation hazards. Urban floods affected 137 of these cities on more than three occasions from 2008 to 2010. From 2012 to 2015, the number of cities in China that suffered urban inundation was 184, 234, 125, and 154, respectively, including major cities like Beijing, Shanghai, Guangzhou, and Shenzhen [8]. Urban inundation increasingly threatens the sustainable development of urban areas.

The traditional approach to flood management emphasizes the design of engineering infrastructures that resist urban floods. However, with the increasingly unpredictable, diverse, and intense weather events that affect cities, a purely physical resilience improvement to tackle urban flooding is apparently inadequate [9]. The notion of individual resilience that occupies the social dimension refers to the behaviors adopted by individuals to cope with and adapt to natural disasters [10]. The environment of a place that underpins the functions or unique memories of people can reshape individuals’ subjective attributes and behaviors [11], leading to their unique way of understanding natural disasters. Hence, the environment and environmental cognition of people should be considered factors influencing individual resilience [12,13]. However, exploration of the dynamics created by different forms of interaction in the complex people–place relationship that influences individual resilience to natural disasters is widely ignored.

Shenzhen is a rapidly urbanizing city with a constant population influx, attracting numerous workers and tenants from rural China [14]. An example of a community often affected by floods is the sub-district of Gongming. Demographically, Gongming is home to numerous rural workers with over half of their dwellings in urban villages, which are of low quality and easily affected by natural disasters [15]. The area is therefore ideal for the exploration of individual resilience among residents, including, but not limited to, poor and vulnerable rural workers. With the case area of Gongming, this research focuses on the following research questions: 

(1) What are the current levels of individual resilience of residents living in different communities that are vulnerable to urban flooding? 

(2) What are the environmental factors affecting individual resilience, and more specifically, how does the environmental cognition of people towards their residence influence individual resilience?

Different residential environmental factors help generate environmental cognitions for people that can influence individual resilience to a certain degree. Accordingly, we hypothesize that (1) local variations exist in the individual resilience of residents in different communities and (2) environment elements comprising the physical form of a residential place correlates with individual resilience. The remainder of this paper consists of seven parts. Following the introduction, the conceptual framework is described in Section 2. The methodology is described in Section 3. The results are presented in Section 4, and their implications are discussed in Section 5. The limitations of this research are summarized in Section 6. Finally, the conclusions are provided in Section 7.

## 2. Conceptual Framework

### 2.1. Conceptualising Individual Resilience from Behavioural Research

This research focuses on individual resilience to address responses to natural disasters. Individual resilience is a multidimensional concept that comprises the self-ability and collective ability of a neighborhood or geographically defined area to enable individuals to cope with and adapt to natural disturbances and efficiently resume their daily lives [16].

Behavioral research provides ways to operationalize resilience by determining whether certain behaviors exhibit resilient thinking [17,18,19]. In terms of urban flooding, preparation for urban flooding is a behavior that demonstrates resilient thinking [20]. Controlling and learning certain knowledge in dealing with urban flooding are also considered resilient behaviors [21]. Urban flooding experiences and learned experiences are positive behaviors that can help people cope with and adapt to urban flooding [22]. In addition, resilient behavior is exhibited by certain habits, such as checking the weather report or early-warning signals of heavy rain [23], or creative and flexible strategies carried out by the residential community and government. These strategies include flexible working and school systems or real-time distribution system on escape pathways, which can benefit people in adapting to urban flooding [24].

### 2.2. Environmental Correlates of Individual Resilience

The environment, especially the residence, is of the greatest importance in one’s life [25]. The activities important to a person’s life are focused in and around the home [26]. The elements of the physical and social environment constitute the material form of a residence, which is an essential feature because this form can distinguish different residences and bring different cognitions in people towards the varying environment [27].

On the one hand, the objective residential environment can influence individual resilience. For example, in the physical environment, natural and manmade drainage networks play a critical role in coping with urban floods [13]. The efficiency and capacity of drainage networks in people’s residences can help deal with heavy rainstorms to prevent urban flooding disasters. In relation to the social environment, a safe neighborhood with less crime and good structure, especially a well-managed and gated commercial property, can always deal with urban flooding and conduct countermeasures more efficiently and methodically compared with urban villages [28,29,30].

On the other hand, the subjective cognitions on the residential environment can influence individual resilience. Existing studies have demonstrated that the form of a place provides psychological and physiological benefits to people by adding motivations to interact with the environment [27]. Environmental cognitions of people can boost or reduce human behaviors that are embedded in social and physical activities. These activities facilitate the social interactions and encounters of residents, which benefits the place to develop social relations. A “relationship-rich” place is conducive to observational learning, where the environment allows people to learn coping and adaptation behaviors from one another [12]. For example, good perceptions on the quality of the physical environment [31], including satisfaction with the environment, garbage disposal, amount of construction, and green space in the local area, contribute to social activities and people engagement [32,33,34]. Individual perceptions of the quality of the social environment, including perceptions of social cohesion and social networks, can also benefit from formal and informal social interactions [35,36]. These environment elements can facilitate the rich relationship of a place, which is a contributing factor for people to learn strategies to cope with and adapt to urban flooding.

## 3. Methodology

In line with the two research questions, our primary task is to evaluate the individual resilience of the selected study area and identify the people–place factors of individual resilience. In particular, we explore the environmental factors of individual resilience on the community level (objective residential environment) and individual level (subjective residential environment).

### 3.1. Study Site

Shenzhen is a populated coastal city in the southeast part of China. Its subtropical maritime monsoon climate brings quite a considerable amount of rainfall. Shenzhen has a long rainfall season from April to September [8]. The Meteorological Bureau of Shenzhen Municipality (2018) reports that the average annual rainfall of Shenzhen in 2018 was 1935.8 mm. Since 2010, precipitation and heavy storms were concentrated and frequently affected Shenzhen. In 2018, eight typhoons entered Shenzhen within 500 km, and four of these caused winds and heavy storms that relentlessly inundated the city. In total, Shenzhen had 134 rainstorm days (37% of the year) [37]. Accordingly, urban flooding was particularly severe, bringing considerable difficulties to residents and even threats to their lives [8].

The selected study site is called Gongming, located in the Guangming New District of Shenzhen, China, within the Maozhou River basin (Figure 1). The total area is 37.68 km^2^, of which 69.8% are impervious surfaces [8]. Gongming includes 11 communities and over 80% of its properties are urban villages. The statistical data from the Shenzhen Water Authority pointed out that Gongming is one of the places that suffers severe urban flooding [38]. This finding presents the need to research the individual resilience of its people and the environmental factors, which are the bases of urban flood mitigation in this area.

### 3.2. Individual Resilience

Behavioral research uses self-reported measures to gauge people’s responses to target situations [39,40]. In this research, individual resilience is evaluated based on the self-reported resilience-oriented strategies adopted by residents and their communities to manage urban flooding.

According to the conceptualization of individual resilience from behavioral research in the second part of this research, residents were asked 15 questions (Table 1) to obtain information on their coping and adaptive behaviors towards urban flooding, including preparations for the potential risk, learning experiences, past coping strategies, habits, and other creative and flexible strategies to adapt to urban flooding. The questionnaire provides raw data that require further processing by transformation (i.e., the conversion of choices into different variables with choices) and theoretical orientation. In this study, a high index value indicates a high level of individual resilience. Respondent behaviors are standardized to a score ranging from 0.00 to 10.00 on two-point, three-point, or four-point Likert scales, which depend on possible and suitable choices for the question. The average of all the selected strategies is the final score of individual resilience, which is then assigned a value (0.00 is lowest, 10.00 is largest) for each person.

### 3.3. Determinants of Individual Resilience

Socio-demographic factors, including gender, age, income, marital status, occupation, and education level, are selected as control variables due to their proven influence on individual resilience [41]. Residents’ self-efficacy and disaster awareness towards urban flooding, such as being confident about overcoming natural disasters, considering such disasters as dangerous, remaining sensibly prepared in dealing with them or possessing previous experience in similar situations [41,42], are also selected to verify existing research in the context of rapidly urbanized rural areas in China.

At the community level, the physical environment resilience from urban flooding is reflected through the drainage capacity. Drainage capacity is an integrated factor that depends on rainfall intensity, flood duration, drainage system efficiency, terrain, land usage, and soil types [43,44,45]. In this study, the precipitation process on May 11 2014 (once in a century) in Gongming was used to simulate the drainage capacity with the Stormwater Management Model (for further details, see Appendix A of Drainage Capacity) on a spatial basis to match the sewerage system data on the same year (2014) for the urban flooding simulation and parameter calibration. This event has led to severe urban flooding in the study area.

In terms of the social environment, measures for urban flooding are reflected through the “gatedness of the community” and “property types”, from which most other factors can be deduced (e.g., non-gated urban villages usually have lower income levels and higher crime rates). The study area has 11 communities (see administrative division map) with a total of 100 residential zones (visual identification based on Google Maps: resolution ratio is 0.0694 (5/72) meter/pixel in Shenzhen) (Figure 1). The gatedness and property types of the 100 residential zones are identified through first-hand observation via a field trip and second-hand data [46]. The measurements of environmental variables at the community level are shown in Table 2 and the calculated results are in Table 3.

In this study, the dependent variable is the score of individual resilience. With our environmental factors covering two levels, namely, individual and community levels, hierarchical linear modeling (HLM) is adopted [47]. HLM is a multilevel analysis method (commonly composed of two levels) that can disentangle the effects of independent variables operating at different spatial scales [47]. In the model, the variables comprising environmental cognitions at the individual level that vary by subject are inputted as Level 1, whereas those comprising the objective environmental attributes of communities are inputted as Level 2.

A total of 24 independent variables are identified from the responses to the questionnaire and are then classified into three groups: control variables at an individual level, objective environmental variables at a community level, and subjective environmental cognitions at an individual level. Among the factors comprising subjective environmental cognitions, three subgroups of explanatory variables can be derived, including self-efficacy and disaster awareness, perception of the quality of the physical environment and perception of the quality of the social environment. To conduct the HLM, these variables are recorded by quantifiable measurements (Table 4).

### 3.4. Participants of the Questionnaire Survey

This study included 973 adults aged 16 (rational men standard) or above who participated on a voluntary basis. The main data collection method was face-to-face surveys at randomly selected locations. The survey period (July 15–October 30, 2016) included weekdays and weekends from 9:00 am to noon and from 3:00 pm to 8:00 pm to ensure that workers and non-workers were surveyed.

Finally, 733 (75.33% of the total number of participants) effective responses were obtained. The sample consisted of 443 males and 290 females (1.5:1 male to female ratio). In the sample, 444 residents were married (60.6%). Three age groups were considered: 16–18 (2.7%), 18–40 (82.7%), and 40 and above (14.6%). One-third of the surveyed residents had an income of no more than 3000 CNY /month and approximately half (53.5%) earned 3000–7000 CNY/month. Nearly two-thirds (65.7%) of the surveyed residents were employed, including part-time workers (5.9%). Two-fifths of the sample needed to work more than five days per week (41.5%), whereas some others worked five days per week (29.7%). A quarter (25.4%) of the surveyed residents were educated up to middle school, and another quarter attained a professional degree or higher education (25.6%).

## 4. Results

### 4.1. Individual Resilience

Table 5 presents the descriptive statistics of individual resilience using the standardized score of the responses to 15 questions provided by the 733 residents in Gongming.

In the study area, the range of individual resilience scores is 9.80, with 0.20 as the lowest score and 10.00 as the highest. The mean score of individual resilience is 4.62, which is the intermediate level. The median is 4.67, reflecting that the individual resilience of over half of the residents is at an intermediate to low level. Compared with a normal distribution, the distribution of the individual resilience values shows a sharp peak and flat tails through the kurtosis value. In addition, the skewness demonstrates several extremely high values on the right. These results indicate that several residents in the study area have basic and effective strategies to overcome urban floods, but the number of residents with intermediate, low or even extremely low levels of individual resilience remains large.

### 4.2. Factors Influencing Individual Resilience

Table 6 shows the explanatory power of all the selected variables with six models. The first five models only consider the factors at the individual level. The sixth model considers factors at the individual and community levels.

The models are listed as follows: Model 1, blank regression model; Model 2, regression model that only considers the control factors; Model 3, regression model considering the control factors and self-efficacy and disaster awareness; Model 4, regression model considering the control factors, self-efficacy and disaster awareness and perception of quality of the physical environment; Model 5, regression model considering the control factors, self-efficacy and disaster awareness, and perceptions of the qualities of the physical and social environments; and Model 6, regression model considering all the above factors at the individual and community levels. 

Model 2 reveals that among the six control variables, gender and education levels are essential to the changes in the level of individual resilience. Model 3 indicates that “perceived leadership during urban flooding disaster”, “sensitivity to urban flooding”, “concerns about children during urban flooding”, “experienced urban flooding” and “losses during urban flooding” are significant factors in the variance of individual resilience. Model 4 suggests that “areas of green lands in or around communities” and “living environment of communities” explain the changes in individual resilience. Model 5 implies that “mutual trust within communities”, “perception of life in your community” and “help and support from community organizations” are significant factors in the changes of individual resilience. Model 6 shows that after including the community-level explanatory variables (i.e., drainage capacity, property types, and gatedness of communities) in the multilevel statistical model, the variance of the random component at the community level decreases substantially. Among the variables that captured variance in the dependent variable, only the “gatedness of communities” is statistically significant (negative) in its correlation with individual resilience.

Combining the measurement of these variables in Table 4 and the coefficients from the regression models (see Table 6), we can identify the following characteristics of residents who are likely to possess high individual resilience.

#### 4.2.1. Leadership

Individuals who perceive leadership during urban flooding disasters are likely to possess high individual resilience in Gongming. This finding is in accordance with similar research abroad [48]. However, the results generate a belief that a social group exists among those who perceive leadership during urban flooding, who claim to be capable of adopting strategies to cope with and adapt to urban flooding in a resilient manner (with high individual resilience) and are overconfident with their leadership during difficulties (leadership). Whether such strategies are genuinely and wholly implemented requires further testing and is discussed in the following section.

#### 4.2.2. Disaster Awareness

A group of residents in the study area appears to be sensitive to urban flooding. Residents with high concerns over children’s safety, particularly those with children, are likely to demonstrate high individual resilience. Residents who have experienced urban flooding and suffered from losses are also more willing to behave in a resilient manner compared with those who have not.

These findings indirectly demonstrate the residents’ weak awareness of the dangers of urban flooding unless they have already been adversely affected. People with high individual resilience are usually passively driven to improve their disaster awareness.

#### 4.2.3. Perceptions of Physical and Social Environments

A good perception of their living environment enables individuals to improve their resilience. Residents with acceptable perceptions of the physical environment, such as a clean environment and abundant green lands in or around communities, are likely to exhibit satisfactory individual resilience.

People with a good perception of the social environment also tend to have high levels of individual resilience. In our research area, the residents who enjoy a sense of well-being in their communities, can receive help and support from their living communities, have friendly neighbors, and possess the willingness to trust them are likely to have high levels of individual resilience.

## 5. Discussion

The main purpose of this study is to test if environmental factors, particularly the environmental cognitions of residents, correlate individual resilience to urban flooding based on self-reported measures of individual resilience and the statistical method of HLM. In the case of Gongming, environmental factors at the community and individual levels significantly influence individual resilience. The following discussion is provided based on the implications of the research findings.

### 5.1. Active Coping and Adaptation to Urban Flooding (People)

In our study area, people with the confidence to lead others out of difficulties demonstrate high individual resilience. “Perceived leadership during urban flooding disaster” (“Leadership” in Table 7) is selected for correlation analyses on the demographic characteristics of selected residents. The models in Table 7 suggest that low-income, less-educated, and single men have reported high confidence to show leadership during difficulties. Their socio-economic characteristics generate the belief that a social group in the study area is overconfident and may be unable to cope with and adapt to urban flooding as reported when real disasters occur. This result reflects a defect in the measurement of individual resilience because such exaggerations by respondents cannot be eliminated. Further refinements are required in future research to remove such bias.

### 5.2. Passive Coping and Adaptation to Urban Flooding (People–Disaster)

Interactions between people and disasters can be passively driven by the former’s personal perception of the latter. Low disaster awareness, including insensitive personal risk perception and little experience of disasters or drills, limits individual resilience. The surveyed residents with high concerns over children’s safety in Gongming are likely to be able to cope with and adapt to urban flooding. By contrast, those who have lost personal possessions in previous urban flooding events are likely to have high individual resilience. In summary, personal perceptions of natural disturbances can be passively driven by personal needs, eventually resulting in resilience-oriented behavior.

### 5.3. Physical Environment and Individual Resilience (People–Place)

The interaction between people and place, and how this relationship affects resilience, is verified as a significant predictor of individual resilience. Although the regression results suggest that variations in the physical environment on a community level cannot distinguish residents with high individual resilience from those with low resilience, the personal perceptions of living environment can influence individual resilience. In Gongming, people who show high individual resilience are satisfied with the quality of the physical environment of their residences. Individuals with a positive perception of their surroundings, such as a clean and tidy physical environment, are likely to have high individual resilience. In addition, residents who believe that their residences have adequate green lands are likely to demonstrate resilience-oriented behaviors. These findings suggest that unlike the drainage capacity of residents’ residences, which is highly related to the occurrences of urban flooding, cognition, and perception of clean, tidy, hygienic, and green living environments are the main drivers for people’s resilience-oriented behaviors. The underlying mechanism may be that the high quality of the physical environment perceived by people can provide psychological and physiological benefits to those living in and around the area [49]. These benefits can lead to many physical and social activities, such as making friends and building social networks, and a high level of fondness towards the place [50].

### 5.4. Social Environment and Individual Resilience (People–Place)

At the individual level, social environment is embedded in residents’ perceptions towards the social cohesion of their residential communities. For Gongming residents, the sense of well-being in their residential areas determines whether social networks can be efficiently built, which will further nurture individual resilience. In addition, residents who believe in and trust their neighbors and community managers always show resilient behaviors. The cognitive social cohesion of their communities explains the changes in the levels of resilience. The social cohesion that connects institutions and organizations across levels and scales facilitate information flow among residents, highlighting the importance of social environments in influencing individual resilience. This conclusion enriches the existing literature in the context of developed countries and proves that the cognitive services provided by communities facilitate human behaviors in developing countries. Efficient and reliable social cohesion within enhanced social environments are the prerequisites of individual resilience, facilitating people to develop coping and adaptation strategies in tandem with high individual resilience.

At the community level, the results demonstrate that the surveyed respondents living in non-gated communities are likely to have high individual resilience. By comparison, residents in gated communities seem to possess relatively low individual resilience. Gated communities in China symbolize good community life and satisfactory urban services [51]. Given that the cognitive quality of the physical and social environment and disaster awareness can influence individual resilience, we conclude that this social group seems dependent on the services provided by the external environment but ignores improvements to their own capacity to cope with and adapt to urban flooding. Specifically, their disaster awareness, passively driven by personal needs, is offset by the high quality of services and resources provided by the external environment, resulting in less resilience-oriented behaviors.

These findings suggest that at present, for residents to enhance their resilience to natural disasters in China, the main driver is the passive compulsion by past dangers and not the active preparation for future potential dangers, especially for those residing in the outskirts of a city. Moreover, a social group was found to be overconfident and may be unable to withstand an urban flooding disaster. This research proves a significant correlation between resilience nurturing and environmental cognition: a better perception of the quality of the living environment indicates a higher individual resilience of the residents. As a mediator, individual cognition of the environment can help nurture and provide a new pathway for enhancing individual resilience. Therefore, urban planning and management strategies, which affect and alter different social groups’ perceptions of the physical and social environments, deserve further exploration in future research for urban managers, scholar, and planners to enhance individual resilience.

## 6. Limitation

One limitation of this study lies in the individual resilience assessment. Subjective bias is inevitable in self-reported methods. Correlation analyses suggest that although residents reported that they have resilience-oriented behaviors, they cannot cope with or adapt to urban flooding when the disaster actually occurs. This misreport reflects a defect in the measurement of individual resilience because such exaggerations by individual respondents cannot be eliminated. Further refinements are required in future research to remove such bias.

In addition, the results of the statistical analyses in this study merely provided possible findings within the research domain. The results of this research incorporate uncertainties, which require an in-depth interview to further clarify the findings.

Perhaps most importantly, this cross-sectional study cannot demonstrate causality. This study builds on experimental research that has, under controlled conditions, proven that effects similar to those observed in this study are produced by the interaction of environmental cognition and individual resilience. However, residents in other types of communities may behave differently. Additional research on other case areas and time series are necessary for further investigation.

## 7. Conclusions

This research adopts a livelihood perspective to provide a creative method of evaluating individual resilience on a relative basis. Environmental factors, embedded into environmental cognitions reported by individuals and environment factors that may influence human behaviors at the community level, are explored as determinants of individual resilience. Half of the residents in Gongming lack individual resilience. This lack of resilience may be attributable to the fact that these residents are already struggling to satisfy their basic living needs and developing social cohesion and disaster awareness. This lack may also be caused by the high quality of services and resources provided to residents in good communities, thereby relaxing their disaster awareness that should further influence their judgment in adopting resilience-oriented behaviors.

Our findings illustrate one explanatory mechanism behind the positive relationships between the living environment and individual resilience, with the individual cognition as the mediator. This outcome supports the theory of environmental cognition and identifies the quality of the social and physical environments that can influence (support or hinder) people’s resilience-oriented behaviors. Most importantly, we identify qualitative and quantitative differences between what are usually considered urban and natural environments that play the same role as the people’s perceptions of urban and natural environments. Instead, attention should be focused on interactions of urban environments with residents, as captured by the case study in Gongming. Strategies should not only design the engineering approach of infrastructure improvement but also integrate human closeness to provide residents with a clean, tidy, hygienic, green, and socially cohesive environment. Under this situation, urban planning should emphasize the perceived importance of individual resilience among these social groups. Such planning strategies, which affect and alter different social groups’ perceptions of their physical and social environments, deserve further exploration in future research for urban managers, scholars, and planners. Such efforts can create resilience-oriented socio-ecological urbanism, enhancing the capacities of residents to respond to natural disasters. Our results demonstrate the utility of the concept of resilience in urban environments and provide an innovative methodological toolbox for their application. We expect similar studies in other regions to reach parallel or contradicting results, increasing our collective knowledge about the complex interplay between the environment and individual resilience in different urban settings from various parts of the world.

## Figures and Tables

**Figure 1 ijerph-16-02559-f001:**
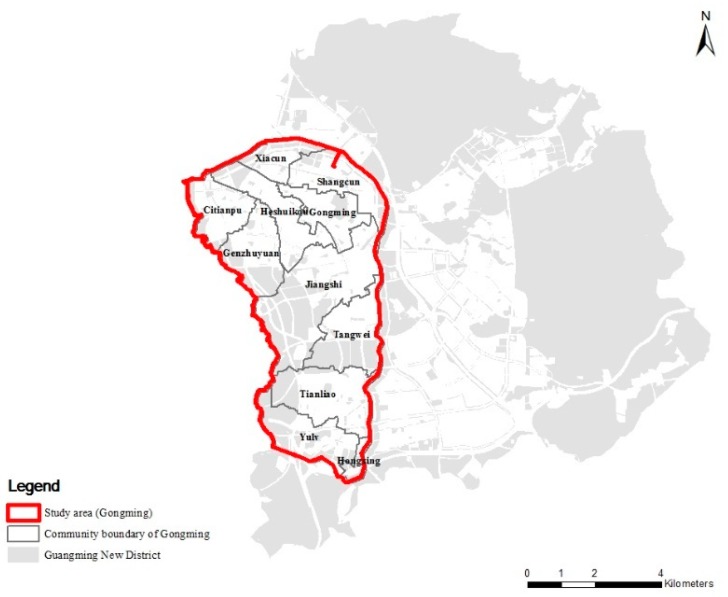
Study area (Gongming).

**Table 1 ijerph-16-02559-t001:** Questions for individual resilience.

Questions
(1) Preparing for the potential urban flooding risk
1. Do you conduct some preparation for heavy rain or urban flooding?
2. Does your living zone have daily maintenance on pipelines and pipe junctions?
3. Does your living zone have commissioners to repair the pipelines after they have been jammed?
(2) Learnt to cope with and adapt to urban flooding
4. Do you know how to help yourself escape from urban flooding?
5. Do you take the initiative to study the knowledge of disaster prevention and risk aversion?
6. Does your living zone provide education for improving your capacity for adapting to disasters?
(3) Past coping strategies during urban flooding
7. Do you know where you can escape from the heavy rain or urban flooding when you are outside?
8. Do you know which areas usually experience urban flooding around your residences?
9. Does your living zone have countermeasures to overcome urban flooding?
(4) Habits to cope with and adapt to urban flooding
10. Do you have the habit to check the weather report?
11. Do you disseminate safe-haven news to your families and friends after you have known it?
12. Do you know the early-warning signal of heavy rain?
(5) Creative and flexible strategies to adapt to urban flooding
13. Do you follow the early-warning signal to arrange your travel flexibly?
14. Do you participate in building new strategies for disaster prevention of your living zone?
15. Does your living zone have innovative strategies for adapting to disasters?

**Table 2 ijerph-16-02559-t002:** Measurements of independent variables (community level).

Attributes of Community Environment	Measurement
Physical environment	
Drainage capacity (ordinal)	Scores: 0.00–10.00
Management	
Property types (ordinal)	Percentage of the commercial residential zones out of the total area of residential zones of a community
Gatedness of communities (ordinal)	Percentage of the gated residential zones out of the total area of residential zones of a community

**Table 3 ijerph-16-02559-t003:** External environmental attributes of each community.

Community	Drainage Capacity * (Standardized Score: 0.00–10.00)	Gatedness of Each Community (Percentage)	Commercial Property of Each Community (Percentage)
Citianpu	7.21	100.00%	0.00%
Genzhuyuan	0.32	100.00%	0.00%
Gongming	5.50	48.31%	21.19%
Heshuikou	4.68	88.96%	33.74%
Hongxing	4.80	84.00%	0.00%
Jiangshi	3.28	78.29%	17.83%
Shangcun	2.65	88.57%	65.71%
Tangwei	3.04	42.86%	28.57%
Tianliao	6.69	15.25%	1.69%
Xiacun	3.44	3.13%	3.34%
Yulv	8.56	97.85%	3.23%
Average	4.56	67.93%	13.04%

Note: * Drainage capacity is the hydraulic capacity and performance of an urban drainage system. Drainage capacity is calculated as the average standardized score (0.00 to 10.00) of 100 residential zones comprising 11 communities. The average score of the residential zones comprising each community is the final drainage capacity.

**Table 4 ijerph-16-02559-t004:** Independent variables and their measurements.

Independent Variables	Measurement
**Control Variables**	
1. Gender	0 = female; 1 = male
2. Marital status	0 = single; 1 = marriage
3. Age	0 = equal or older than 41; 1 = equal or younger than 40
4. Income	0 = equal or more than 3000 CNY/month; 1 = lower than 3000 CNY/month
5. Occupation	0 = no work; 1 = fulltime/part-time job
6. Education level	0 = equal or higher than professional education level; 1 = lower than professional education level
**External Variables at Community Level**	
Physical environment	
7. Drainage capacity	Scores: 0–10
Social environment	
8. Property types	Percentage of the commercial residential zones out of the total number of residential zones of a community (0–100%)
9. Gatedness of communities	Percentage of the gated residential zones out of the total number of residential zones of a community (0–100%)
**Internal Variables on Individual Level**	
Self-efficacy and disaster awareness	
10. Perceived leadership during urban flooding disaster	0 = no; 1 = yes
11. Feelings about urban flooding	1 = no dangerous; 2 = not very dangerous; 3 = dangerous; 4 = very dangerous
12. Sensitivity to urban flooding	0 = no; 1 = yes
13. Concerns about children during urban flooding	0 = no; 1 = yes
14. Concerns about the elderly during urban flooding	0 = no; 1 = yes
15. Experienced urban flooding	0 = no; 1 = yes
16. Previous losses during urban flooding	1 = none; 2 = not too much; 3 = general; 4 = much; 5 = very much
Perception of quality of the physical environment	
17. Amount of construction projects in progress around communities	1 = quite a lot; 2 = general; 3 = not too many
18. Areas of green lands in or around communities	0 = not too many; 1 = a lot
19. Living environment of communities	1 = very good; 2 = good; 3 = general; 4 = bad; 5 = very bad
20. Cleanliness and hygiene of communities	1 = very clean; 2 = clean; 3 = general; 4 = not clean; 5 = very unclean
Perception of quality of the social environment	
21. Mutual trust within communities	0 = do not have; 1 = have
22. Mutual help within communities	0 = do not have; 1 = have
23. Perception of life in the community	1 = very bad; 2 = bad; 3 = general; 4 = good; 5 = very good
24. Help and support from community organizations	0 = do not have; 1 = have

**Table 5 ijerph-16-02559-t005:** Descriptive statistics of individual resilience in Gongming (*n* = 733).

Descriptives	Statistics	Individual Resilience
Range	Statistic	9.80
Minimum	Statistic	0.20
Maximum	Statistic	10.00
Mean	Statistic	4.62
Median	Statistic	4.67
Standard Deviation	Statistic	1.67
Variance	Statistic	2.79
Skewness	Statistic	0.27
	Standard Error	0.09
Kurtosis	Statistic	0.08
	Standard Error	0.18

**Table 6 ijerph-16-02559-t006:** Estimations of the determinants of individual resilience.

Independent Variables	Model 1		Model 2		Model 3			Model 4		Model 5		Model 6	
	Parameters	SE	Parameters	SE	Parameters		SE	Parameters	SE	Parameters	SE	Parameters	SE
**Fixed Effects**													
Individual level													
Intercept	4.614 ***	0.084	4.403 ***	0.191	3.397 ***	0.388		4.952 ***	0.466	2.479 **	0.677	2.793 **	0.693
Gender			0.436 ***	0.096	0.400 **	0.127		0.444 ***	0.122	0.430 ***	0.120	0.421 ***	0.121
Marital status			0.421 ***	0.121	0.182	0.134		0.126	0.130	0.135	0.127	0.133	0.128
Age			−0.006	0.100	−0.020	0.178		−0.035	0.171	−0.014	0.168	0.004	0.168
Income			−0.097	0.152	−0.079	0.159		−0.054	0.154	−0.068	0.152	−0.066	0.151
Occupation			−0.103	0.127	−0.135	0.154		−0.171	0.149	−0.190	0.146	−0.179	0.146
Education level			−0.264 *	0.139	−0.263	0.133		−0.259	0.128 *	−0.293 *	0.126	−0.294 *	0.126
Perceived leadership during urban flooding disaster					0.623 **	0.123		0.564	0.119 ***	0.516 ***	0.117	0.521 ***	0.117
Feelings about urban flooding					−0.091 ***	0.088		−0.058	0.085	−0.026	0.083	−0.030	0.083
Sensitivity to urban flooding					0.151	0.129		0.185	0.125	0.192	0.122	0.210 *	0.123
Concerns about children during urban flooding					0.400 *	0.161		0.327	0.155 *	0.315 *	0.152	0.298 *	0.152
Concerns about the elderly during urban flooding					0.207	0.157		0.122	0.151	0.065	0.149	0.115	0.150
Experienced urban flooding					0.433 ***	0.124		0.445	0.122 ***	0.389**	0.120	0.393 ***	0.120
Previous losses during urban flooding					0.224 ***	0.061		0.213	0.058 ***	0.165 **	0.058	0.163 **	0.058
Amount of construction projects in progress around communities								−0.119	0.080	−0.125	0.078	−0.115	0.080
Areas of green lands in or around communities								0.430	0.117 ***	0.373 **	0.116	0.381 ***	0.116
Living environment of communities								−0.529	0.099 ***	−0.415 ***	0.101	−0.387 ***	0.102
Cleanliness and hygiene of communities								−0.025	0.096	0.039	0.096	0.041	0.095
Mutual trust within communities										0.514 *	0.214	0.484 *	0.214
Mutual help within communities										0.277	0.266	0.326	0.266
Perception of life in the community										0.409 ***	0.093	0.403 ***	0.093
Help and support from community organizations										0.723 *	0.298	0.687 *	0.299
Community level													
Drainage capacity												−0.032	0.031
Property types												0.194	0.381
Gatedness of communities												−0.474 *	0.214
**Random Effects**													
var (U_0_)	0.0350 *		0.0250 *		0.0450 *			0.0090		0.0080		0.0003	
ρ (U_0_)	0.0125		0.0092		0.0181			0.0040		0.0037		0.0001	

Note: SE means standard error. Dependent variable: Individual resilience. The total number of surveyed individuals is 733. The number of surveyed communities is 11. var (U_00_) is the variance component at the community level. ρ (U_0_) is the proportion of variance at the community level to the total variance. * *p* < 0.1, ** *p* < 0.01, *** *p* < 0.001.

**Table 7 ijerph-16-02559-t007:** Correlations between socio-demographic factors and perceived leadership during urban flooding disasters.

Item	Gender	Marital Status	Age	Occupation	Shenzhen Hukou	Education Level	Housing Ownership	Housing Types	Income
Leadership	0.100 **	0.308 **	0.134 **	−0.051	−0.026	0.011	0.031	−0.088 *	0.02

Note: * *p* < 0.05, ** *p* < 0.01, *** *p* < 0.001.

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
