# Peer review of "Linkage Between the Environment and Individual Resilience to Urban Flooding: A Case Study of Shenzhen, China"

_ijerph, 2019, doi:10.3390/ijerph16142559_

Round 1

Reviewer 1 Report

TITLE

Instead of using the word “association” which could mean a connection between people or organizations or group of people organized for a joint purpose I would suggest to use the word “linkage”, which could mean the relationship between two things, especially where one affects the other.

ABSTRACT

Abstract is very clear, written with good English. It follows a logical structure and catches the attention of the reader by clearly explaining the problem, how this study has addressed it and it introduces a hint of the results obtained.

INTRODUCTION

Page 2 Line 11: “that” should be added between disasters and occupies.

Page 2 Line 13: “that” should be added between place and underpinned.

Page 2 Line 19: reference needed

Page 2 Line 21: reference needed

Page 2 Line 25: The word “following” should be added in line 25 between the and research.

Page 2 Lines 26-29 should be presented in bullet points

Page 2 Lines 27: The word “determinants” should be changed with “factors”

CONCEPTUAL FRAMEWORK

This section should be shortened and incorporated within the literature. Some concepts in fact are described a few times within the two different sections and this does not help the flux of the paper.

METHODOLOGY

Page 3 Line 45: “Correlates” should be changed with “factors”

Table 2: Unites drainage capacity??

Line 7-8 page 6: This concept is not clearly explained. Needs rephrasing

Section 3.3: Not written in good English. Needs to be checked.

Line 9 page 7: brackets need to be closed (60.6)

Not clear what table 4 presents and which is the individual resilience

It is a bit confusing which methodology was used and how the variables were selected. The reader if filled with single statements which seem to all have an influence on the results but instead of being connected they are fragmented and not clear.

RESULTS

Table 5: what are the 6 models?

RESULTS

A lot of variables are introduced but it is difficult to scale up this case study and see what would have been the implications of different initial parameters.  It is good to see a section stating the limitations but it would very interesting to have the opinion of the authors on incorporating uncertainty within their results.

Authors defined two research questions but do not identify environmental elements affecting individual residence. In line 45 Page 13, they mention environmental factors and write “especially environmental cognition of residents” but the latest is not an environmental factor.

Author Response

Referee 1

1.        TITLE

Instead of using the word “association” which could mean a connection between people or organizations or group of people organized for a joint purpose I would suggest to use the word “linkage”, which could mean the relationship between two things, especially where one affects the other.

Re: In the revised manuscript, we have used ‘linkage’ instead of ‘association’. See the change on line 2 of page 1.

2.        ABSTRACT

Abstract is very clear, written with good English. It follows a logical structure and catches the attention of the reader by clearly explaining the problem, how this study has addressed it and it introduces a hint of the results obtained.

Re: Thank you for your kind summary.

3.        INTRODUCTION

Page 2 Line 11: “that” should be added between disasters and occupies.

Page 2 Line 13: “that” should be added between place and underpinned.

Page 2 Line 19: reference needed

Page 2 Line 21: reference needed

Page 2 Line 25: The word “following” should be added in line 25 between the and research.

Page 2 Lines 26-29 should be presented in bullet points

Page 2 Lines 27: The word “determinants” should be changed with “factors”

Re: We acknowledge that your comments are all necessary. In this submission, we have revised the Introduction following your suggestions.

4.        CONCEPTUAL FRAMEWORK

This section should be shortened and incorporated within the literature. Some concepts in fact are described a few times within the two different sections and this does not help the flux of the paper.

Re: We have revised the conceptual framework to make it shorter and easy to follow. Please find the revised parts on pages 3 to 4.

5.        METHODOLOGY

Page 3 Line 45: “Correlates” should be changed with “factors”

Re: We have revised the word on the basis of your suggestion.

Table 2: Unites drainage capacity??

Re: The unit is the standardized score of drainage capacity. In particular, the final drainage capacity is calculated as the average standardized score (0.00 to 10.00) of 100 residential zones that comprise 11 communities. The average score of the residential zones comprising each community is the final drainage capacity. We have marked this change as a note in the figure. Please see lines 16-18 of page 7.

Line 7-8 page 6: This concept is not clearly explained. Needs rephrasing

Re: We have revised and improved the expression following your suggestions. Please check the explanation on lines 3-17 of page 8.

Section 3.3: Not written in good English. Needs to be checked.

Re: We have improved the writing and copyedited the manuscript. In addition, we have found a professional company to proofread the entire paper so as to improve the accuracy of the English expression of this study.

Line 9 page 7: brackets need to be closed (60.6)

Re: We have revised the word following your suggestion (line 9 of page 9).

Not clear what table 4 presents and which is the individual resilience

Re: Individual resilience is evaluated based on 15 questions that are shown and mentioned in the methodology part of this research (line 13 of page 5). Based on this evaluation, the score of individual resilience for each person can be calculated and the descriptive statistics of individual resilience in Gongming (n = 733) can be found in table 5 of the revised paper.

It is a bit confusing which methodology was used and how the variables were selected. The reader if filled with single statements which seem to all have an influence on the results but instead of being connected they are fragmented and not clear.

Re: This may be a problem caused by our poor English expression. We have improved the expression for the methodology section and found a native English editor to proofread the manuscript. We hope the revisions for the methodology can now meet your requirements.

6.        RESULTS

Table 5: what are the 6 models?

Re: We have explained the six models on lines 12-23 of page 10 in the revised manuscript.

7.        RESULTS

A lot of variables are introduced but it is difficult to scale up this case study and see what would have been the implications of different initial parameters.  It is good to see a section stating the limitations but it would very interesting to have the opinion of the authors on incorporating uncertainty within their results.

Re: We have added the limitation you mentioned on lines 36-39 of page 16 in the revised manuscript.

Authors defined two research questions but do not identify environmental elements affecting individual residence. In line 45 Page 13, they mention environmental factors and write “especially environmental cognition of residents” but the latest is not an environmental factor.

Re: In this research, we focused more on the person–environment transactions and how the transactions influence the nurturing of individual resilience. Our research has two levels of environmental factors. The first is the objective environmental factors on the community level and the second comprise the factors that demonstrate the subjective environmental cognitions on an individual level. The regression models confirm our viewpoints that the environmental cognition of residents is the main factor explaining the degree of individual resilience compared with the objective environmental factors.

Reviewer 2 Report

In general

The article explores how individual resilience of residents can be enhanced to better prepare them for natural disasters, in particular urban flooding, and establishes a theoretical framework for the concept of individual resilience to urban flooding. The study offers an approach for analyzing the key factors that limit individual resilience from the environmental perspective, thereby providing a basis for formulating corresponding policy recommendations to improve resilience effectively through urban planning and management strategies. This is illustrated by a case study, by which the authors concluded that the lack of individual resilience to urban flooding is mainly attributable to few green spaces, low quality of the built environment, mutual distrust and not well-being perceived by residents. At the community level, the results suggest that the social environment is pivotal to individual resilience.

Comments

Some passages are difficult to read because of grammatical errors. I suggest that a native English speaker revises the text.

Question

On the one hand, people with a good perception of the social environment also tend to have high levels of individual resilience. On the other hand, people living in gated communities seem to possess relatively low individual resilience. These two statements seem to be contradictory. Can the authors clarify this contradiction?

Author Response

Referee 2

Comments and Suggestions for Authors

In general

The article explores how individual resilience of residents can be enhanced to better prepare them for natural disasters, in particular urban flooding, and establishes a theoretical framework for the concept of individual resilience to urban flooding. The study offers an approach for analyzing the key factors that limit individual resilience from the environmental perspective, thereby providing a basis for formulating corresponding policy recommendations to improve resilience effectively through urban planning and management strategies. This is illustrated by a case study, by which the authors concluded that the lack of individual resilience to urban flooding is mainly attributable to few green spaces, low quality of the built environment, mutual distrust and not well-being perceived by residents. At the community level, the results suggest that the social environment is pivotal to individual resilience.

1.        Comments

Some passages are difficult to read because of grammatical errors. I suggest that a native English speaker revises the text.

Re: We have improved the writing and copyedited the manuscript. In addition, we have found a professional company to proofread the entire paper so as to improve the accuracy of the English expression of this study.

2.        Question

On the one hand, people with a good perception of the social environment also tend to have high levels of individual resilience. On the other hand, people living in gated communities seem to possess relatively low individual resilience. These two statements seem to be contradictory. Can the authors clarify this contradiction?

Re: Our research has two levels of environmental factors. The first is the objective environmental factors on the community level and the second comprise the factors that demonstrate the subjective environmental cognitions on an individual level. The regression models confirm our viewpoints that the environmental cognitions of residents are the main factors explaining the degree of individual resilience compared with the objective environmental factors. The regression results in this research show that people with a good perception of the social environment (subjective environmental cognitions) tend to have high levels of individual resilience. However, on the objective level, the regressions show that people living in gated communities seem to possess relatively low individual resilience. We have explained this on lines 3-13 of page 16. At the community level, the results demonstrate that the surveyed respondents living in non-gated communities are likely to have high individual resilience. By comparison, residents in gated communities seem to possess relatively low individual resilience. Gated communities in China symbolize good community life and satisfactory urban services (Pow and Kong, 2007). Given that the cognitive quality of the physical and social environments and disaster awareness can influence individual resilience, we conclude that this social group seems dependent on the services provided by the external environment but ignores improvements to their own capacity to cope with and adapt to urban flooding. Specifically, their disaster awareness, passively driven by personal needs, is offset by the high quality of services and resources provided by the external environment, resulting in less resilience-oriented behaviours.

Pow CP and Kong L (2007) Marketing the Chinese dream home: Gated communities and representations of the good life in (post-) socialist Shanghai. Urban Geography 28(2): 129-159.

Reviewer 3 Report

Comments to the authors:

The authors show in this paper an interesting work to analyse the key factors that limit individual resilience from an environmental perspective. The manuscript is well-presented and it is easy to read, although some spelling or typing errors can be found.

Therefore, I think that this manuscript can be published after minor revision (text editing).

Author Response

Referee 3

Comments and Suggestions for Authors

Comments to the authors:

The authors show in this paper an interesting work to analyse the key factors that limit individual resilience from an environmental perspective. The manuscript is well-presented and it is easy to read, although some spelling or typing errors can be found.

Therefore, I think that this manuscript can be published after minor revision (text editing).

Re: Thank you for your affirmation of this research. We have improved the writing and copyedited the manuscript. In addition, we have found a professional company to proofread the entire paper so as to improve the accuracy of the English expression of this study.

Reviewer 4 Report

After reading your manuscript “Association between environment and individual resilience to urban flooding: A case study of Shenzhen, China”, I highlight next MAJOR comments:

Abstract. This section cannot exceed 200 words. 

Introduction.Definition of resilience is required. Preparedness, planning, absorption, recovery and adaptation are abilities to be considered when referring to resilience concept (National Academy of Sciences (2012). Disaster Resilience: A National Imperative. The National Academies Press). The linkage between resilience and environment/ individual resilience should be revealed. In order to emphasize the relevance of urban flooding with respect to natural disasters, some statistics should be given. In the same vein, further data should support the selection of Gongming as case study. Arguments to back up hypotheses of lines 29 to 31 (page 2) should be included. A final paragraph depicting shortly the structure of the manuscript is highly recommended.

Conceptual framework. This section is confusing. Concepts of environmental/ individual resilience should be clearly determined, as well as their attributes in relation to psychological / physiological aspects mentioned by authors.For instance, line 36 is unclear. 

Methodology.The use of different Likert scale (P4L12) is very controversial, since they can cause the lack of scores consistency by affecting respondents´ evaluation criteria. The organization of information shown in Table 1 is unclear. Basis to formulate the 15 questions covering individual resilience are unknown. Different score ranges should determine individual resilience performance levels rather than a continuum of scale from 0 to 10. The characterization of physical environment through the sole consideration of the drainage capacity is very simplistic and inaccurate (P5 L1-2). Same view for social environment aspect (P5L7-8). Gender, age, income and so can vary, so they cannot be considered as control variables (P4L19) (control variables are constant and unchanged throughout the course of the investigation). The hypothesis reflected in P5L15-16 assuming that “the sample of each residential zone is the unbiased estimation of the true population of each residential zone” is unacceptable. Thus, both assumptions are fatal, impacting very negatively in the research. Criteria/sources to rate “gatedness of the community” and “commercial property of each community” in Table 2 are unknown. Reasons why the precipitation process on 11 May 2014 was deemed to simulate drainage capacity were omitted (P5L2-3). The correlation between many of the 24 independent variables of Table 3 and questions addressed in Table 1 is very unclear. 

General Comments. References must be numbered in order of appearance in the text rather than the citation format used by authors. Proof reading is necessary. 

On the whole, the study revealed critical flaws derived from several assumptions made by authors that cast into great doubts the validity of the work. 

Author Response

Referee 4

Comments and Suggestions for Authors

After reading your manuscript “Association between environment and individual resilience to urban flooding: A case study of Shenzhen, China”, I highlight next MAJOR comments:

1.        Abstract. This section cannot exceed 200 words.

Re: In the revised manuscript, we have shortened the Abstract to achieve conciseness and readability (lines 13-31of page 1).

2.        Introduction. Definition of resilience is required. Preparedness, planning, absorption, recovery and adaptation are abilities to be considered when referring to resilience concept (National Academy of Sciences (2012). Disaster Resilience: A National Imperative. The National Academies Press). The linkage between resilience and environment/ individual resilience should be revealed. In order to emphasize the relevance of urban flooding with respect to natural disasters, some statistics should be given. In the same vein, further data should support the selection of Gongming as case study. Arguments to back up hypotheses of lines 29 to 31 (page 2) should be included. A final paragraph depicting shortly the structure of the manuscript is highly recommended.

Re: We acknowledge that your comments are all necessary. In this submission, we have revised the Introduction following your suggestions and marked the changes in red.

3.        Conceptual framework. This section is confusing. Concepts of environmental/ individual resilience should be clearly determined, as well as their attributes in relation to psychological / physiological aspects mentioned by authors. For instance, line 36 is unclear.

Re: We acknowledge that your comments are all necessary. In this submission, we have revised the Conceptual framework following your suggestions and marked the changes in red.

4.        Methodology.

The use of different Likert scale (P4L12) is very controversial, since they can cause the lack of scores consistency by affecting respondents´ evaluation criteria. Different score ranges should determine individual resilience performance levels rather than a continuum of scale from 0 to 10.

Re: We admit that different Likert scales are applied in this research, which may influence the research results to a certain degree. However, numerous published papers have applied similar methods to score the targeted topic, such as Reynolds (2010), Zenger et al. (2013). Moreover, in this research, we standardized the score of each question into a 10-point system, which is an acceptable method to manage questionnaires with different numbers of selections comprising the targeted topic (Hartley, 2014).

Hartley, J., 2014. Some thoughts on Likert-type scales. International journal of clinical and health psychology14(1), pp.83-86.

Reynolds, C.R., 2010. Behavior assessment system for children. The Corsini encyclopedia of psychology, pp.1-2.

Zenger, M., Finck, C., Zanon, C., Jimenez, W., Singer, S. and Hinz, A., 2013. Evaluation of the Latin American version of the Life orientation Test-Revised. International Journal of Clinical and Health Psychology13(3), pp.243-252.

The organization of information shown in Table 1 is unclear. Basis to formulate the 15 questions covering individual resilience are unknown. Criteria/sources to rate “gatedness of the community” and “commercial property of each community” in Table 2 are unknown.

Re: The organization and structure of this section have several problems that lead to the misunderstanding of the selection of questions comprising individual resilience evaluation. Situations are similar to the criteria/sources to rate ‘gatedness of the community’ and ‘commercial property of each community’. In fact, the selection criteria are shown in the conceptual framework section that we have marked in blue (lines 2 to 7, 9 to 25 of page 4).

The characterization of physical environment through the sole consideration of the drainage capacity is very simplistic and inaccurate (P5 L1-2). Same view for social environment aspect (P5L7-8).

Re: We evaluate the drainage capacity of the study area by considering the land use types, geographical factors (such as slope, height, and so on), and sewerage system density and quality (lines 14 to 16 of page 6). It is a compulsory factor covering the main physical environment of the research area. We do not want to repeat the environmental elements and therefore used one integrated factor to express the physical environment. This was similarly done for the social environment factor. The social environment for urban flooding on the community scale is reflected through the ‘gatedness of the community’ and ‘property types’, from which most other factors could be deduced (e.g. non-gated urban villages usually have lower income levels and higher crime rates) (lines 5 to 7 of page 7).

Gender, age, income and so can vary, so they cannot be considered as control variables (P4L19) (control variables are constant and unchanged throughout the course of the investigation).

Re: We acknowledge that in our research, several socio-demographic factors are closely related to the dependent variable mainly due to their contributing effect. However, these are not the focal factors in this research, and therefore we control them to find the targeted factors and their relationships with our dependent variables.

The hypothesis reflected in P5L15-16 assuming that “the sample of each residential zone is the unbiased estimation of the true population of each residential zone” is unacceptable. Thus, both assumptions are fatal, impacting very negatively in the research.

Re: We apologize for our poor English expression that misunderstood your comment. In truth, we selected our samples strictly on the basis of the female and male properties and age stages according to the yearbook of Gongming, Shenzhen. It is not an assumption of the research. The relevant sentence has been removed to improve that section. The correct expression is on page 7 (lines 7-14).

Reasons why the precipitation process on 11 May 2014 was deemed to simulate drainage capacity were omitted (P5L2-3).

Re: We have added the selection reason as follows: ‘In this study, the precipitation process on May 11 2014 (once-in-a-century) in Gongming was used to simulate the drainage capacity with the Stormwater Management Model (for further details, see Supplementary Analysis of Drainage Capacity) on a spatial basis to match the sewerage system data on the same year (2014) for the urban flooding simulation and parameter calibration. This event has led to severe urban flooding in the study area.’  See lines 1-3 of page 7.

The correlation between many of the 24 independent variables of Table 3 and questions addressed in Table 1 is very unclear.

Re: We have explained their correlations in the conceptual framework section. In addition, we have revised and improved the English expression to make it easier to follow (lines 20-22 of page 9).

5.        General Comments. References must be numbered in order of appearance in the text rather than the citation format used by authors. Proof reading is necessary.

Re: In the revised manuscript, we have proofread the entire paper and revised the reference style following your suggestion.

6.        On the whole, the study revealed critical flaws derived from several assumptions made by authors that cast into great doubts the validity of the work.

Re: In the revised manuscript, we have exerted our best efforts to overcome the stated disadvantages, and we hope you can reconsider to accept our research.

Round 2

Reviewer 4 Report

After reading the revised manuscript “Linkage between environment and individual resilience to urban flooding: A case study of Shenzhen, China”, I highlight next MAJOR comments:

Abstract. Lines 3-4 are wrong. Please revise the wording. 

Introduction.The definition of resilience was changed in P2L6-9, but the citation remains the same as the original manuscript (Davoudi, 2012).The linkage between resilience and environment/ individual resilience should be strengthened by an extensive literature review. Grounds to select Gongming as case study should be provided (P5 L8-10 are very vague). Strong hypotheses of lines 8 to 11 (page 3) should be underpinned. 

Conceptual framework. This section is confusing because of the introduction of several concepts without a clear linkage. Environment should be precisely contextualized. Subsection 2.2. is mostly oriented to housing environment which significantly constraints the scope of the study as reflected in the article title. Objective / subjective factors should be determined. 

Methodology.It was not disclosed a whole overview of stages followed in the research. Although behaviors were scored within a range of 0 to 10 points, the use of two-point, three-point or four-point Likert scales distorts respondents´ valuation given that assessment threshold are different, influencing their appraisals.  Basis to formulate the 15 questions covering individual resilience remain unknown. The characterization of physical environment through the sole consideration of the drainage capacity is very simplistic and inaccurate. Same view for social environment aspect (P7L6). Gender, age, income and so can vary, so they cannot be considered as control variables (P6L6-8) (control variables are constant and unchanged throughout the course of the investigation). Sources of “gatedness of the community” and “commercial property of each community” in Table 3 are unknown. Solid reasons why the precipitation process on 11 May 2014 was deemed to simulate drainage capacity were not given (P7L18). The correlation between many of the 24 independent variables of Table 4 and questions addressed in Table 1 is very unclear. 

In relation to the survey, although it was undertaken face to face, most selected participants (82.7%) were in an age range from 18 to 40, whilst by 60% were male and the rest female. This situation may significantly bias the findings. Furthermore, the “randomly selected locations” (P9L4) from the 11 Gongming communities were not identified. 

On the whole, the revised manuscript revealed that the majority of prior comments remain unaddressed.